# TBP-Related Factor 2 as a Trigger for Robertsonian Translocations and Speciation

**DOI:** 10.3390/ijms21228871

**Published:** 2020-11-23

**Authors:** Roman O. Cherezov, Julia E. Vorontsova, Olga B. Simonova

**Affiliations:** Koltzov Institute of Developmental Biology, Russian Academy of Sciences, Vavilova str. 26, 119991 Moscow, Russia; ro-tcherezov@yandex.ru (R.O.C.); vjul83@mail.ru (J.E.V.)

**Keywords:** centric fusion, nondisjunction, pericentromeric regions, chromocenter, TATA-box-binding protein-related factor, D1 chromosomal protein, *Drosophila*

## Abstract

Robertsonian (centric-fusion) translocation is the form of chromosomal translocation in which two long arms of acrocentric chromosomes are fused to form one metacentric. These translocations reduce the number of chromosomes while preserving existing genes and are considered to contribute to speciation. We asked whether hypomorphic mutations in genes that disrupt the formation of pericentromeric regions could lead to centric fusion. *TBP-related factor* 2 (*Trf2*) encodes an alternative general transcription factor. A decrease of TRF2 expression disrupts the structure of the pericentromeric regions and prevents their association into chromocenter. We revealed several centric fusions in two lines of *Drosophila melanogaster* with weak *Trf2* alleles in genetic experiments. We performed an RNAi-mediated knock-down of *Trf2* in *Drosophila* and *S2* cells and demonstrated that Trf2 upregulates expression of *D1*—one of the major genes responsible for chromocenter formation and nuclear integrity in *Drosophila*. Our data, for the first time, indicate that Trf2 may be involved in transcription program responsible for structuring of pericentromeric regions and may contribute to new karyotypes formation in particular by promoting centric fusion. Insight into the molecular mechanisms of Trf2 function and its new targets in different tissues will contribute to our understanding of its phenomenon.

## 1. Introduction

Centric-fusion translocation (a Robertsonian translocation) is a type of chromosome rearrangement, in which two long arms of acrocentric chromosomes fuse to form one metacentric [1]. Centric chromosome fusion often accompanies pathological genetically inherited human syndromes (trisomy 13 (Patau) syndrome and trisomy 21 (Down) syndrome), so it is important to study the source of their occurrence. Such chromosomal rearrangements, like others, can be caused by environmental risk factors (for example, ionizing radiation or chemical carcinogens), but they also may be caused by mutations that destabilize the compact structure of chromatin in pericentromeric regions [2]. It has been suggested that centric fusion may cause changes in the number of chromosomes in the karyotype of species, and thus play a role in speciation and influence the course of evolution in general [3,4,5,6,7,8,9,10]. This is possible due to the unique feature of such rearrangements not to disrupt the total number of genes (which can be fatal for the organism), but to change the number of chromosomes, contributing to the genetic isolation of the population—a condition necessary for new species to evolve [11,12]. In this regard, the search for genes whose mutations may be able to influence the course of the evolutionary process through a change in the chromosome set is of fundamental biological importance. Mutations of such genes should create favorable conditions for the emergence of centric fusion, in other words, they should affect the structure of heterochromatin in pericentromeric regions. Obviously, such mutants will have an increased rate of chromosome nondisjunction compared to karyotypically normal animals. Indeed, the proper segregation of sister chromatids between daughter cells requires coordinated interaction of centrosomes, centromeres, kinetochores, spindle fibrils, topoisomerases, proteolytic processes, and motor proteins [13,14]. A number of genes control each steps of this process. Mutations in these genes lead to chromosome nondisjunction and tumorigenesis [15]. However, it is unlikely that they will contribute to centric fusion. On the other hand, chromosomes must be “prepared” (or structurally organized) when they enter meiosis (or mitosis). Structural disorganization of chromosomes or their regions that control the correct pairing of homologues during meiosis often results in the incorrect chromosome segregation. We believe that it is more likely that mutations in genes that disrupt the organization of pericentromeric regions will promote centric fusion of chromosomes.

One way of eukaryotic genome organization is the formation of a chromocenter, which is evolutionally involved in the regulation of chromosome behavior in dividing cells not only among insects but also among plants, mammals, mollusks, and even yeast [16,17,18,19,20,21]. This nuclear structure arises in differentiated somatic and germ cells during interphase and meiotic prophase and is generated by the association of pericentromeric regions of all or separate groups of chromosomes. Studies on *Drosophila* and mice have clearly demonstrated that its disorganization leads to genomic disbalance. It has been suggested that the chromocenter performs coordination of chromosome before division and indirectly increases the frequency of crossing over when its forming is delayed [22,23,24]. A detailed study of chromocenters in *Drosophila* oocytes has shown that their structure is genetically programmed [25,26].

In screening for genes that control the formation and reorganization of chromocenter, the high frequency of chromosome nondisjunction is the main peculiarity in the progeny of mutant parents. Moreover, recent studies on *Drosophila* and mouse cells have shown that satellite DNA in pericentromeric regions plays a critical role in maintaining genome integrity in the cell nucleus since depletions of two sequence-specific satellite DNA-binding proteins, D1 and Prod, disrupts chromocenters and leads to the formation of micronuclei [7,27]. Nevertheless, it is still not known whether mutations that disrupt chromocenter formation and, in particular, pericentromeric heterochromatin, can cause centric fusion.

In *Drosophila*, *TBP-related factor 2* (*Trf2*) encodes an alternative general transcription factor that is homologous to vertebrate Trf2 protein and belongs to a conservative TATA box-binding protein (Tbp) gene family [28]. The ubiquitous presence of Trf2 allows studying its functions in *Drosophila*. Previously discovered *lawc^p1^* (*leg-arista-wing complex*) mutation appeared to be the only viable mutation that decreases Trf2 expression in *D. melanogaster* [29,30]. It was demonstrated that *lawc^p1^* suppresses the phenotype of flies with mutations in genes that encode Polycomb Group proteins, which are negative epigenetic regulators of transcription via chromatin modification [31]. At the same time, it was found that *lawc^p1^* increase the effect of transvection (or allelic complementation [32]) caused by disruptions of the homologous chromosome pairing and suppress the position effect variegation. A decrease in *Trf2* gene expression in mice and *Drosophila* could result in the disruption of chromatin condensation [33,34]. Genetic experiments on *Drosophila* with the use of mass crosses demonstrated a high frequency of chromosome nondisjunction in *Trf2* mutant female meiosis and this anomaly accompanied with an abnormal chromatin packaging and particularly with the failure of chromocenter organization [35]. Taking into account the specific *Trf2* mutant phenotype associated with disruption of pericentromeric heterochromatin, we hypothesized that even a slight decrease in *Trf2* gene activity could cause a chromosome centric fusion.

To test our hypothesis, new genetic screening of flies with X chromosome nondisjunction and attached-X chromosomes in lines with *Trf2* lethal mutations were performed. We revealed several flies carrying attached X in two lines with weak *Trf2* alleles. We discovered that these events more often occurred in meiosis of females with lower levels of nondisjunction. We also demonstrated that Trf2 upregulates expression of *D1*—one of two major genes responsible for chromocenter formation and nuclear integrity in *Drosophila*. Our data, for the first time, indicate that Trf2 may be involved in a transcription program responsible for structuring of pericentromeric regions and contributing to new karyotypes formation in particular by promoting centric fusion.

## 2. Results

### 2.1. Frequency of X Chromosome Nondisjunction and Existence of Attached-X Chromosomes in Females with Lethal Trf2 Mutations

Earlier, in genetic experiments on *D. melanogaster* using mass crosses, we demonstrated a high frequency of X chromosome nondisjunction in females with *Trf2* lethal mutants [35]. However, mass crosses do not allow identifying females with attached-X chromosomes. To provide this we performed individual crosses of about 30 females from seven lines that contain *Trf2* lethal mutations. Further, phenotypic analysis of F1 progeny was carried out to determine their belonging to normal or exceptional classes.

In the case of normal chromosome segregation in female meiosis, only regular classes will be present in F1 generation (Figure 1A). In the case of X chromosome nondisjunction in females during meiosis, splitting of traits will be observed in their F1 generation, and both regular and exceptional classes will be present (Figure 1B). Attached-X chromosomes consist of two full-length X chromosomes sharing a common centromere so that they are always inherited together. In crosses of normal males to females carrying attached-X chromosomes, male progeny will inherit their X from their father and their Y from their mother (X^X/Y). It means that the only exceptional flies will be found in the offspring if the parent female carries attached X (Figure 1C). By following this principle, we could determine if the parent female had chromosome nondisjunction or if the female was carrying attached-X chromosomes.

Phenotypic analysis revealed that in seven lines tested, two, (*l(1)G0425* and *l(1)G0356*), exhibited the presence of females with attached X (Table 1 and Appendix A). Their male progeny had the expected patroclinous phenotype (“yellow body”).

To confirm the presence of attached-X chromosomes (X^X/Y) in discovered *l(1)G0425* and *l(1)G0356* females, they were bred in individual crosses with *yw*/Y males and metaphase chromosome plates of larval neuroblasts were prepared (Figure 2). The mitotic karyotype of *D. melanogaster* females comprises four pairs of chromosomes (2*n* = 8), three autosomal pairs plus the pair of X chromosomes (Figure 2A). Two pairs of metacentric autosomes II and III form four cruciform-like structures. One pair of small acrocentric autosomes IV forms two dot-like structures and one pair of large submetacentric X chromosomes forms two V-like structures (Figure 2B). Metaphase figures of *l(1)G0425* and *l(1)G0356* females with attached-X do not show V-like structures, but demonstrate five cruciform-like structures (two pairs of autosomes II, III and one X^X) plus J-shaped Y-chromosome (Figure 2C,D).

We cannot strictly assert that there were no such females in other five lines, since the selection of females for analysis was carried out randomly. Perhaps, if we increase the sample of analyzed females, we will be able to find such unique individuals. Nevertheless, our experiments demonstrate that a decrease in *Trf2* gene expression may indeed lead to such a phenomenon as chromosome centric fusion.

Then, we carried out a total count of females and males for each phenotypic class and calculated the frequency of X chromosome nondisjunction. A range of strengths was seen among the various alleles. The maximal levels of X chromosome nondisjunction were in three *Trf2* alleles (*l(1)G0166*, *l(1)G0152*, l*(1)EF520*) which exceeds the frequency of nondisjunction in controls by approximately 20 times. In total, Four alleles (*l(1)G0425*, *l(1)G0356*, *l(1)G0424*, *l(1)G0295*) were weak, giving levels of 7.18%, 11.59% 17.39%, and 14,51%, respectively (Table 1, Appendix A, Figure 3A).

We also calculated the frequency of parent females, which produced exceptional progeny in tested lines (Figure 3B). It turned out that in four of seven lines (*l(1)G0425*, *l(1)G0356*, *(1)G0424*, *l(1)G0295*) about half (40–48.15%) of females have X chromosome nondisjunction. It is interesting that these were lines with flies bearing weaker alleles. In two lines (*l(1)G0166*, *l(1)G0152*) only a quarter (20% and 26.66%) of females have X chromosome nondisjunction and in the *l(1)EF520* line, 6.67% of females produced exceptional progeny. The last three alleles demonstrated high levels of X chromosome nondisjunction. Thus, we found a negative correlation between the level of chromosome nondisjunction and the number of females producing exclusive offspring in different lines with *Trf2* lethal mutation. Moreover, we have demonstrated that decrease of *Trf2* gene expression may provide chromosome centric fusion.

### 2.2. Decrease of Trf2 Gene Expression Downregulates D1 Chromosomal Protein

Earlier, we demonstrated that the depletion of Trf2 leads to a failure of chromocenter formation and chromatin condensation [35]. On the other hand, chromocenter splitting assumes the disruption of interchromosomal ectopic contacts in the pericentromeric heterochromatin region. *D1 chromosomal protein* (*D1*) encodes a multi-AT-hook chromosomal protein that associates with AT-rich satellites, including the SAT-III repeats of the X chromosome. The depletion of D1 leads to chromocenter disruption and micronuclei formation [7]. As Trf2 encodes the general transcription factor, we suggested that it could drive *D1* gene expression. 

It was previously demonstrated that Trf2-containing complexes selectively recognize TATA-less promoters and initiate not TATA-, but DPE- or TCT-depending transcriptions [36]. To evaluate the potential role of Trf2 in the regulation of *D1* gene transcription we analyzed *D1* promoter regions using bioinformatics approach as described in Materials and Methods. According to Flybase release FB2020_04, the *Drosophila D1* gene encodes three transcripts and has two alternative promoters. We revealed that there are several initiators (Inr) and Bridges (a bipartite core promoter element BridgeI and BridgeII) within predicted promoters in 5′-flanking regulatory regions of *D1* transcripts and only one TATA-box and TFIIB Recognition element (BRE) at the 3′-end -termini of *D1-RC* splice variant of the first exon (Figure 4 and Appendix A). It indicates that Trf2 could be a good candidate to initiate *D1* transcription.

To confirm the ability of Trf2 to affect *D1* gene expression in *Drosophila* we carried out RNAi-mediated knock-down of Trf2 using DNA-directed RNA interference (ddRNAi) technique and quantified the levels of *D1* mRNA before and after Trf2 knock-down. For this, we designed *UAS-Tris* flies with inducible expression of double-stranded RNA homologous to the *Trf2* mRNA. To drive *UAS-Tris* transgene in salivary glands we used *sgs-Gal4* strain. The comparison of the levels of mRNA synthesis by qRT-PCR in salivary glands of *UAS-Tris/+* larvae with *UAS-Tris/+*; *sgs-Gal4/+* larvae demonstrated a decrease in *D1* gene expression confirming that Trf2 may control *D1* expression (Figure 5A). Similar results were obtained when we examine *lawc^p1^/l(1)EF520* and *lawc^p1^/Df(1)RA2* mutant flies with decreased expression of *Trf2* (Figure 5B,C). Mutants with downregulated *Trf2* expression had a drop in *D1* gene expression. Next, we performed RNAi-mediated silencing of *Trf2* in *Drosophila* S2 cells as described in Materials and Methods and confirmed our results (Figure 5D).

In the next experiments, we performed RNAi-mediated silencing of *Trf2* and examined *Drosophila* S2 cells with depleted TRF2. We revealed an increased frequency of the micronuclei formation and the defective reformation of the nuclear envelopes (Figure 6). These defects were similar to those described for *D1* mutant cells [7].

Together our results suggest that mutant *Trf2* phenotype associated with genomic disbalance could be mediated in particular by decreased *D1* gene expression.

## 3. Discussion

Here, in genetic experiments, we have demonstrated for the first time that the decreased expression of *Trf2* could facilitate the linkage of two X chromosomes to the centromere. Attached X in *Drosophila* is a special case of the centric fusion (also referred to as Robertsonian translocation), which involves two homologous acrocentric chromosomes. We were looking for individuals with attached X because in *D. melanogaster* only two—a large X and fourth dot chromosome—from four chromosomes are acrocentric and it is more suitable to search for attached sex chromosomes bearing marker mutations. In other words, we have documented the formation of isochromosomes that carry identical copies of the same arm joined through a single centromere. We do not rule out centric fusion between X and the fourth chromosomes in our experiments, but we have not directly tested this. Nevertheless, we observed the formation of micronuclei in S2 cells with depleted Trf2, which may be a consequence of aneuploidy and lagging chromosome formation in mitosis against the background of translocations between non-homologous chromosomes [37].

We also found a high frequency of X chromosome nondisjunction in females with different *Trf2* alleles. By the strength of nondisjunction, we have divided them into two groups. In lines with a relatively rare appearance of females with nondisjunction of X chromosome (up to 26%), the levels of nondisjunction were the highest, reaching 29% (the first group: *l(1)G0166*, *l(1)G0152*, *l(1)EF520*; Figure 3). Vice versa: in lines with frequent appearance of females with X chromosome nondisjunction (40–48.15%) the levels of nondisjunction were lower, barely reaching 17% (the second group: *l(1)G0425, l(1)G0356, l(1)G0424, l(1)G0295*; Figure 3). However, in all mutant *Trf2* alleles, the frequency of X chromosome nondisjunction significantly exceeded the control one calculated for line with balancer *FM7a* chromosome, on which *Trf2* lethals were maintained (1.4%), and for the line with wild type *Trf2* allele but with the same (as in *Trf2* mutants) genetic background (1.5%). The rarity of females with a high frequency of X chromosome nondisjunction (from the first group) can be explained by the fact that in these lines the effect of *Trf2* mutation on chromosome destabilization is probably stronger than in others, and many gametes die during female meiosis, dropping out of the analysis. Lower levels of nondisjunction in the remaining *Trf2* alleles allows for a larger number of females producing exceptional progeny to survive increasing the possibility of their detection. It is important to note that X-attachments occurred in the second group of alleles. We believe that only weak *Trf2* alleles are able to create optimal “boundary” conditions of genome destabilization, which allow exceptional progeny to survive, including those with centric fusion.

Previously, it was shown that segregation errors observed in *Trf2* mutant meiosis could be a consequence of split chromocenters and improper chromatin condensation not only in *Drosophila*, but also in mice [34,35]. Moreover, immunostaining of *Drosophila* polytene chromosomes revealed that Trf2 resides in pericentromeric regions [38]. However, despite such attractive localization it is unlikely that general transcription factor Trf2 directly participates in pericentromeric heterochromatin organization. Indeed, a recent study demonstrated that Trf2 interacts with TFIIA-L paralog Moonshine and can be recruited to the heterochromatin to initiate a heterochromatin-dependent transcription within PIWI-interacting RNA (piRNA) clusters [39]. This small RNA pathway acts in reproductive cells to prevent active transcription and transpositions of different mobile elements through heterochromatin formation at their location—mostly in pericentromeric region [40]. Thus, the localization of Trf2 in pericentromeric heterochromatin could be explained by its transcriptional activity within piRNA clusters in gonads. Nevertheless, the disorganized chromocenters observed in somatic cells mutant for *Trf2* suggests its indirect participation in the maintenance of high order chromatin structure in pericentromeric region.

It was recently reported that the multi-AT-hook satellite DNA-binding protein, D1, is responsible for packaging of pericentromeric satellite DNA from heterologous chromosomes into chromocenters [7,27]. The bioinformatics analysis of *D1* 5′-flanking regulatory regions allows us to conclude that Trf2 is the most preferred candidate for *D1* transcription activation. The reported observation that Trf2 is enriched at the *D1* gene promoter region supports this possibility [36]. Knock down experiments performed in vivo in *Drosophila* and in S2 cell culture as far as analyses of *Trf2* mutants have clearly demonstrated that the depletion of Trf2 reduces *D1* RNA levels (Figure 5).

Finally, we revealed that RNAi-mediated silencing of Trf2 in *Drosophila* S2 cells leads to prominent defect in nuclear architecture—the formation of micronuclei. This defect may by a consequence of incorrect chromosome segregation induced by different chromosome rearrangements (mostly translocations) [41]. On the other hand, segregation errors lead to discoordination in the reformation of the nuclear envelope [42]. Indeed, we observed Lamin-negative nuclei in Trf2 depleted cells (Figure 6) which indicates a problem with nuclear envelope reformation in cells with decreased Trf2 expression. Cells with loss of *D1* function demonstrate similar anomaly [7] confirming our assumption that Trf2 is the prevalent general transcription factor upregulating *D1* gene expression.

Recent findings revealed the involvement of Trf2 in specialized transcription programs during development [43]. It was postulated that Trf2 facilitated the evolution of the third germ layer (mesoderm) and bilateria [44]. Our data, for the first time, indicate that Trf2 may also be involved in transcription program responsible for the structuring of pericentromeric regions and may trigger new karyotypes formation in particular by promoting centric fusion. Insight into the molecular mechanisms of Trf2 function and its new targets in different tissues will contribute to our understanding of its phenomenon.

## 4. Materials and Methods

### 4.1. Fly Stocks, Rearing Conditions and Genetic Crosses

Flies were kept at 25 °C on standard *Drosophila* wheat meal–yeast–sugar–agar medium. Individual crosses were performed in standard glass vials with 2 to 3 males and 1 female per vial.

The following stocks of *D. melanogaster* with lethal mutations in *Trf2* were obtained from Bloomington Drosophila Stock Centre: *l(1)G0425/FM7a*; *l(1)G0356/FM7a*; *l(1)G0424/FM7a*; *l(1)G0295/FM7a*; *l(1)G0166/FM7a*; *l(1)G0152/FM7a*; and *l(1)EF520/FM7a*. These flies carry lethal *Trf2* mutations induced by *p{lacW}* transposon [45]. *l(1)EF520* produces truncated Trf2 protein (Simonova, personal communication).

X-linked *Trf2* lethal mutations are balanced by *In(1)FM*. This chromosome carries a dominant *Bar* (*B*) marker mutation (narrow eyes) and recessive allele of *yellow* (*y*) gene (yellow body). We crossed *y^+^l(1)/In(1)FM*, *yB* female with males that carried the X chromosome marked by the *y*^1^ mutation (*y*^1^/Y) individually in order to identify exceptional progeny and estimate the frequency of X chromosome nondisjunction and possible presence of attached-X chromosomes in parent females. The regular progeny were *In(1)FM*, *yB*/Y males with narrow eyes and yellow body and 2 classes of females including (1) *In(1)FM*, *yB/yB^+^* (yellow body, kidney-shaped eyes) and (2) *y^+^l(1)/yB*^+^ (grey body and normal oval eyes).

When X chromosome nondisjunction occurred, males and females of exceptional classes (that always differ by phenotypes) were detected. These were X/0 males with normal oval eyes and yellow bodies and XX/Y females with grey bodies and kidney-shaped eyes. Males of the normal class hemizygous for the X-linked lethal allele—*l(1)*/Y—die. Exceptional Y/0 males also die and XX/X super-females had low viability and died at an early age. Therefore, the frequency of X chromosome nondisjunction (Qn) was calculated according to the formula Qn = 100% × 2(X0 + XXY)/(XX + 2XY + 2X0 + 2XXY), where X0 and XXY are the number of flies of exceptional classes; XX and XY are the number of flies of normal classes [46]. The sum of exceptional classes in the numerator was multiplied by 2 in order to take into account lethal classes with the XX/X and Y/0 genotypes. The number of XY males in denominator was multiplied by 2 in order to take into account the class of lethal *l(1)*/Y males.

To calculate the frequency of parental females with nondisjunction (Qf), the following formula was used: Qf = 100% × Nn/N, where Nn is the number of females with X chromosome nondisjunction, N is the total number of analyzed females.

To detect the influence of *In(1)FM* balancer chromosome on the frequency of X chromosome nondisjunction and compare it with the frequency of nondisjunction Qn calculated for tested alleles a control experiment was performed. For this we crossed *In(1)FM*, *B/In(1)FM*, *B* females with *y*^1^/Y males. *In(1)FM*, *B/y*^1^ females of regular class must have kidney-shaped eyes in the F1 progeny caused by a combination of one copy of the *Bar* mutant allele with one copy of the wild-type allele of this gene. *In(1)FM*, *B*/Y males of regular class must have narrow eyes caused by the presence of one copy of the *Bar* mutant allele. Exceptional *In(1)FM*, *B/In(1)FM*, *B*/Y females must have narrow eyes caused by two copies of the *Bar* mutant allele, while *y*^1^/0 males must have normal oval eyes and yellow bodies.

To determine the influence of *p{lacW}* transposon on X chromosome nondisjunction in *Trf2* alleles and to take into account the genetic background other control experiment was performed. We detected the frequency of sex chromosome nondisjunction in *l(1)G0071* line with lethal mutation caused by the integration of the *p{lacW}* transposon not to *Trf2* gene region.

If the parent female produced only exceptional progeny, it was concluded that she had attached X.

The *lawc^p^*^1^ mutation was described earlier [29]. *Df(1)RA2/FM4, y31dsc8dmB* (RA2) flies contain a deletion of [7D10;8A4-5] cytological region which overlaps *Trf2* locus [7E]. *l(1)EF520/lawc^p1^* and *Df(1)RA2/lawc^p1^* heterozygous flies were used in RT-PCR experiments.

The *sgs-GAL4* line was a gift from Dr. Vladik Mogila (Institute of Gene Biology Russian Academy of Science).

### 4.2. Drosophila Melanogaster Schneider 2 (S2) Cell Culture

*Drosophila* S2 cells (ATTC, Rockville, MD, USA) were cultured in Schneider’s *Drosophila* medium (Sigma-Aldrich, St. Louis, MO, USA) supplemented with 10% fetal bovine serum (Thermo Fisher Scientific, Waltham, MA, USA) at 25 °C.

### 4.3. RNA Interference (RNAi) in Cultured Drosophila S2 Cells

RNAi was carried out according to [47]. Cells were pelleted and then resuspended in Schneider’s Drosophila medium at a concentration of 1 × 10^6^ cells/mL. 1 mL of cells suspension per well was seeded in a 6-well culture dish (Corning Costar, Cambridge, MA, USA). To perform RNAi, 15 µg of double-stranded RNA (dsRNA) were added to S2 cell culture. After incubation for 1 h at 25 °C, 2 mL of Schneider’s *Drosophila* medium supplemented with 10% fetal bovine serum were added to each well. Control cells were treated with 15 µg of dsRNA corresponding to eGFP gene fragment that has no homology to *Drosophila* genes. Cells were grown for 72 h at 25 °C and then harvested for RNA extraction. To minimize off-target effects sequence specific for TRF2 RNAi was chosen by using a web-based tool SnapDragon (https://www.flyrnai.org/snapdragon) [48].

### 4.4. Production of dsRNA for RNA Interference in Cultured Drosophila S2 Cells

Sequences used as templates for dsRNA synthesis in in vitro transcription reactions were amplified by polymerase chain reaction (PCR). The primers used for PCR harbored T7 RNA polymerase promoter sequence (TAATACGACTCACTATAGGGAGA) at their 5′-termini followed by gene-specific sequence. The PCR products were purified by using QIAquick Gel Extraction Kit (Qiagen, Valencia, CA, USA). To produce dsRNA corresponding to the genes of interest in vitro transcription was performed by using Invitrogen MEGAscript™ T7 Transcription Kit (Thermo Fisher Scientific, Waltham, MA, USA) in accordance with the manufacturer’s instructions. After the completion of the reactions, dsRNA was precipitated by adding 1/10 volume of 3 M ammonium acetate and 2.5 vol of 100% ethanol, and incubating for 30 min at −70 °C. The dsRNA pellets were air-dried and resuspended to a final concentration of 3 µg/mL and analyzed by 1% agarose gel electrophoresis to ensure the size and integrity of dsRNA. For the amplification by PCR of *Trf2* gene open reading frame fragment, cDNA reversely transcribed from total RNA extracted from S2 cells was used. For the amplification of eGFP open reading frame fragment pEGFP-C1 vector from Clontech (Clontech, Mountain View, CA, USA) was used. The sense and antisense gene-specific sequences were as follows: Trf2 sense 5′-CTCGTTTCCTCAACTTTCGC-3′, Trf2 antisense 5′-CATCTGTTTCAGACGAGGCA-3′; eGFP sense 5′-TAAACGGCCACAAGTTC-3′, eGFP antisense 5′-GTGTTCTGCTGGTAGTGG-3′.

### 4.5. RNA extraction and cDNA Preparation

Total RNA from adult flies, salivary glands, and S2-cultured cells was extracted using RNAzol RT reagent (MRC, Cincinnati, OH, USA) according to the manufacturer’s specifications. To remove genomic DNA contamination RNA samples were treated with TURBO DNase (TURBO DNA-free kit, Invitrogen, Thermo Fisher Scientific, Waltham, MA, USA) according to the manufacturer’s protocol. cDNA was synthesized from 1 µg of total RNA using MMLV RT kit with random priming (Evrogen, Moscow, Russia).

### 4.6. Real-Time Quantitative Reverse Transcription PCR (qRT-PCR)

The qRT-PCR analyses were carried out using qPCRmix-HS LowROX and qPCRmix-HS SYBR+LowROX mixes (Evrogen, Moscow, Russia) according to the manufacturer’s protocol on ABI Prism 7500 Sequence Detection System (Applied Biosystems, Foster City, CA, USA). Differences in gene expression levels were determined using the relative quantification (2^−ΔΔCt^) method [49]. *Rpl32* gene was used as the endogenous control. The levels of mRNA expression of the *Rpl32* and *Trf2* genes were measured using Taqman probe-based chemistry. The levels of mRNA of *D1* gene was measured SYBR-based chemistry with dissociation curve analysis. By optimization of SYBR Green method, its performance and quality could be comparable to TaqMan method [50,51]. The following primer pairs and probes were used for amplification: for *Rpl32* gene primers *Rpl32* sense 5′-CCAGCATACAGGCCCAAGATC-3′, *Rpl32* antisense 5′-ACGCACTCTGTTGTCGATACC-3′, TaqMan probe—FAM- CGCACCAAGCACTTCATCCGCCAC-BHQ1; for *Trf2* gene primers *Trf2* sense 5′-GCTTGCCGCATTTAAACTAAACTC-3′, *Trf2* antisense 5′- TTTGGCTCTTGATTTTGTTGTTGC-3′, *Trf2* TaqMan probe FAM- AGCCTGGACACCGAAGCGAAAAGC–BHQ1; for *D1* gene primers *D1* sense 5′-AGAAGCACGAGGACAATGAC-3′, *D1* antisense 5′-TCTTTGGACACCTTGCCAG-3′. All reactions were carried out in triplicate. The significance of differences in gene expression levels between samples was evaluated by using REST software (Qiagen, Valencia, CA, USA) based on pair wise fixed reallocation randomization test [52]. P-value of less than 0.05 was considered significant.

### 4.7. DNA-Directed RNA-Interference

In vivo RNAi-mediated knockdown of *Trf2* by using DNA-directed RNA-interference (ddRNAi) was accomplished using the *Gal4/UAS* system [53,54]. To create *Tris* plasmid for knock-down of *Trf2* a 577 bp and 525 bp cDNA fragments corresponding to *Trf2* open reading frame were amplified by PCR using primers harboring restriction sites at their 5′-termini and cloned with head-to-head orientation into the *Drosophila* transformation *pUAST* vector [54] to form an inverted repeat. This inverted repeat expressed under control of the *UAS* element [54] as a stem-loop sequence that trigger RNAi-mediated knock-down of *Trf2* gene. The following primer were used for amplification by PCR 577bp and 525bp *Trf2* cDNA fragments, respectively, (restriction sites are underlined): Trf2RI sense 5′-CCGAATTCCTCGTTTCCTCAACTTTCGC-3′, Trf2XhoI sense 5′-CCCTCGAGCATCTGTTTCAGACGAGGCA-3′, Trf2XbaI antisense 5′-CCTCTAGACTCGTTTCCTCAACTTTCGC-3′, Trf2XhoI antisense 5′- CCCTCGAGGGTGCTGATGTTTGGTCATTCG-5′. Structure of *Tris* construct was verified by PCR amplification, DNA sequencing and restriction mapping.

The construct was injected into *w1118* preblastoderm embryos as described elsewhere [55]. Using inverse PCR, we found only one copy of *UAS-Tris* transgenic construct that was located 539 bp upstream of *CG4781* 5′-termini on chromosome 2R (Appendix A). To corroborate that observed phenotype after *Trf2* knock-down was caused only by *Trf2* itself, we performed rescue experiment using *Drosophila* line overexpressing Trf2 protein as described in our earlier work [56].

We used *sgs-GAL4* flies as a driver to induce the expression of the *UAS-Tris* construct in salivary glands [57].

### 4.8. Computational Analysis of Promoters Regions

Berkeley Drosophila Genome Project (BDGP) neural network promoter prediction software (https://www.fruitfly.org/seq_tools/promoter.html) [58] with default parameters was used for *D1* promoter and TSS prediction. Elements Navigation Tool (ElemeNT) (http://lifefaculty.biu.ac.il/gershon-tamar/index.php/element-description/elementv2) [59] with default parameters was used for analysis of core promoter elements. For promoter prediction and analysis, we used sequence from +200 of NM_169261.3 and NM_079562.3 5′-termini to −300 bp of NM_169262.2 5′-termini. All sequences were obtained from NCBI.

### 4.9. Chromosome Cytology, Immunofluorescence Staining and Microscopy

To analyze attached-X chromosomes in metaphases, brains from third-instar larvae were dissected in saline (NaCl 0.7%) and incubated for 1 h in saline with colcemid (Calbiochem, La Jolla, CA, USA) (0.1 µg/mL). Brains were then treated for 8 min with hypotonic solution (0.5% Na citrate), fixed in 4% paraformaldehyde in 45% acetic acid (1:1) for 10 min, and squashed in 45% acetic acid under a 20 × 20 mm coverslip. Chromosome squashes were frozen in liquid nitrogen. After flipping off the coverslip, slides were washed with PBS, incubated for 5 min in SytoxGreen (1:500, Thermo Fisher Scientific, Waltham, MA, USA), washed with cold PBS, and then mounted in Vectashield H-1000 mounting medium (Vector Laboratories, Burlingame, CA, USA) to reduce fluorescence fading.

S2 cells were fixed in 4% paraformaldehyde for 10 min, washed 3 times in PBS and then permeabilized by incubating with 0.2% Triton X-100 in PBS for 5 min at room temperature. After blocking, cells were incubated overnight with the primary mouse monoclonal antibody anti-Lamin Dm (ADL84, Developmental Studies Hybridoma Bank, Iowa City, Iowa, USA) diluted 1:40 in PBS. Secondary antimouse antibodies (1:100) were conjugated to Cy5 (Thermo Fisher Scientific, Waltham, MA, USA). DNA was stained with SytoxGreen (1:500, Thermo Fisher Scientific, Waltham, MA, USA). The preparations were mounted in Vectashield mounting medium (H-1000, Vector laboratories, Burlingame, CA, USA). The resulting immunofluorescence staining were examined using Leica TCS SP5 confocal microscope. Images were processed using Leica LAS X Lite and Adobe Photoshop software.

## Figures and Tables

**Figure 1 ijms-21-08871-f001:**
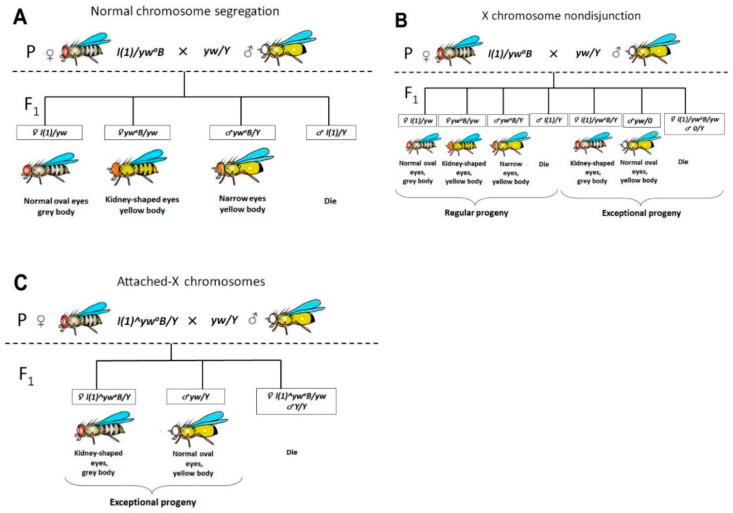
Schematic representation of individual genetic crosses for identifying females with attached-X and females with X chromosome nondisjunction. (**A**) Normal chromosome segregation; (**B**) F1 progeny from a female with X chromosome nondisjunction; (**C**) F1 progeny from female carrying attached-X.

**Figure 2 ijms-21-08871-f002:**
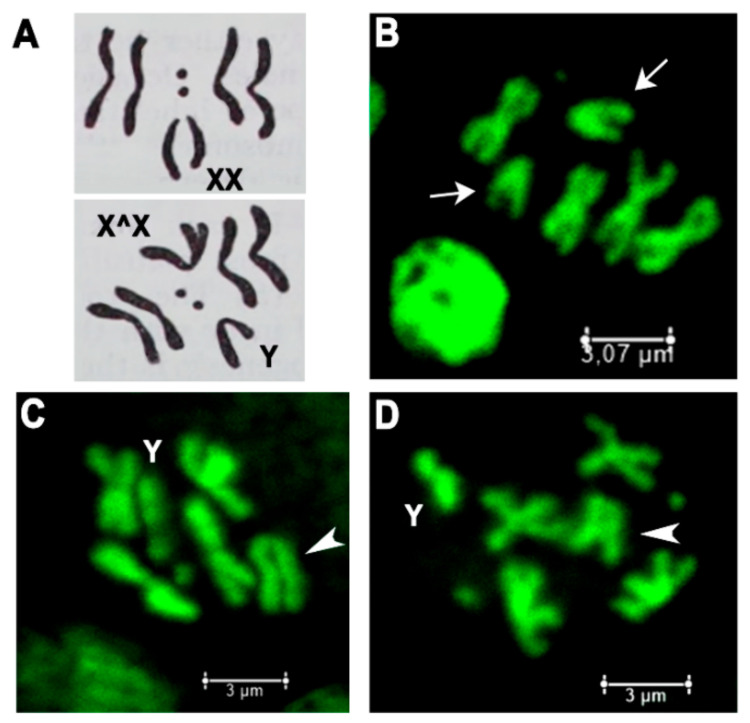
Females with *Trf2* lethal mutations have attached-X chromosomes. (**A**) Schemes of wild type female metaphase plate carrying normal X (upper panel) and attached-X and Y (lower panel). (**B**–**D**) Metaphase plates of larval neuroblasts: (**B**) wild type female; (**C**) *l(1)G0425^FM7a/Y*; (**D**) *l(1)G0356^FM7a/Y*. Arrows point to normal X chromosomes, arrowheads point to attached-X chromosomes.

**Figure 3 ijms-21-08871-f003:**
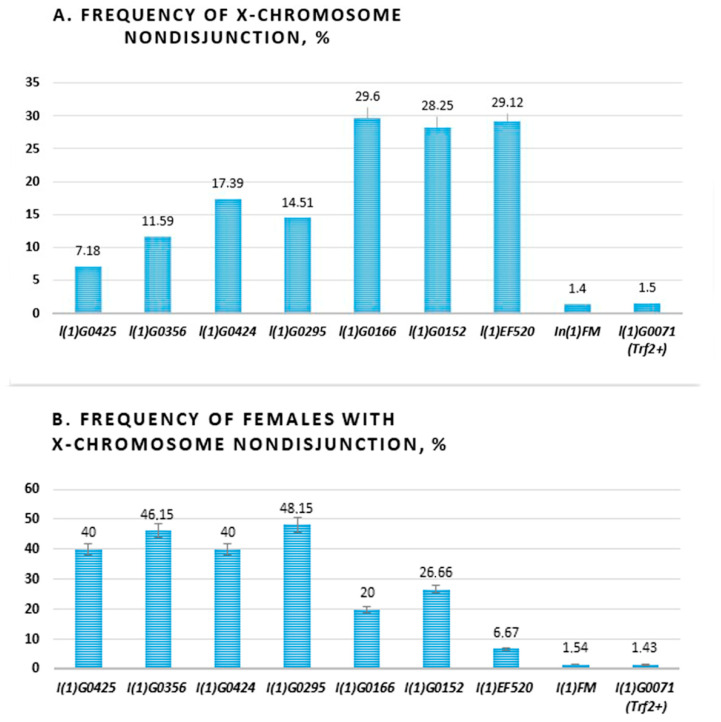
Levels of X chromosome nondisjunction in females with different *Trf2* lethal mutations. (**A**) Levels of X chromosome nondisjunction. (**B**) The frequency of females with X chromosome nondisjunction. Below are alleles of *Trf2*: (*l(1)G0425*, *l(1)G0356*, *l(1)G0424*, *l(1)G0295*, *l(1)G0166*, *l(1)G0152*, *l(1)EF520*), and two controls: balancer *In(1)FM* and *l(1)G0071* with wild type *Trf2* allele.

**Figure 4 ijms-21-08871-f004:**
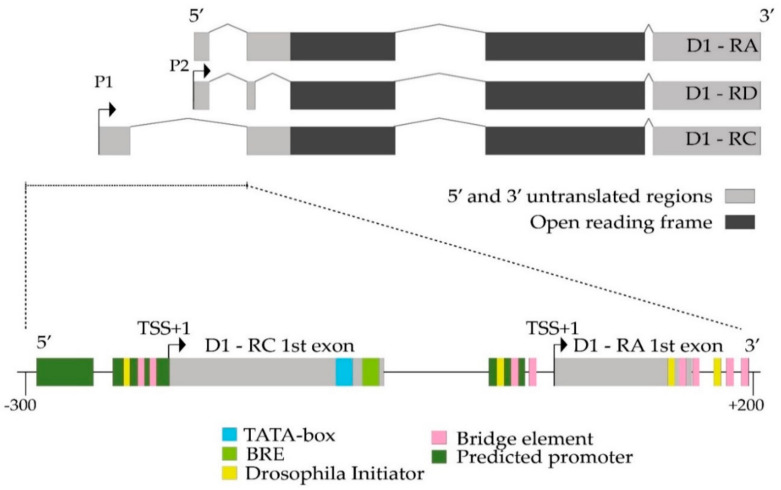
Analysis of putative promoter regulatory elements in *Drosophila melanogaster D1* gene promoter region. (Upper panel) schematic *D1* gene structure, showing the position and approximate size of exons, introns, and open reading frame in different splice variants according to Flybase. The sequence chosen for bioinformatic analysis performed as described in materials and methods is underlined. Two different promoters involved in transcription regulation of *D1* gene designated as P1 and P2. (Lower panel) schematic results of bioinformatic analysis of putative promoter regulatory elements in *D1* promoter. TSS+1—transcription start site (TSS). −300—the distance upstream of D1—RC TSS. +200—the distance downstream of D1—RA and D1—RD TSS. BRE—the B recognition element.

**Figure 5 ijms-21-08871-f005:**
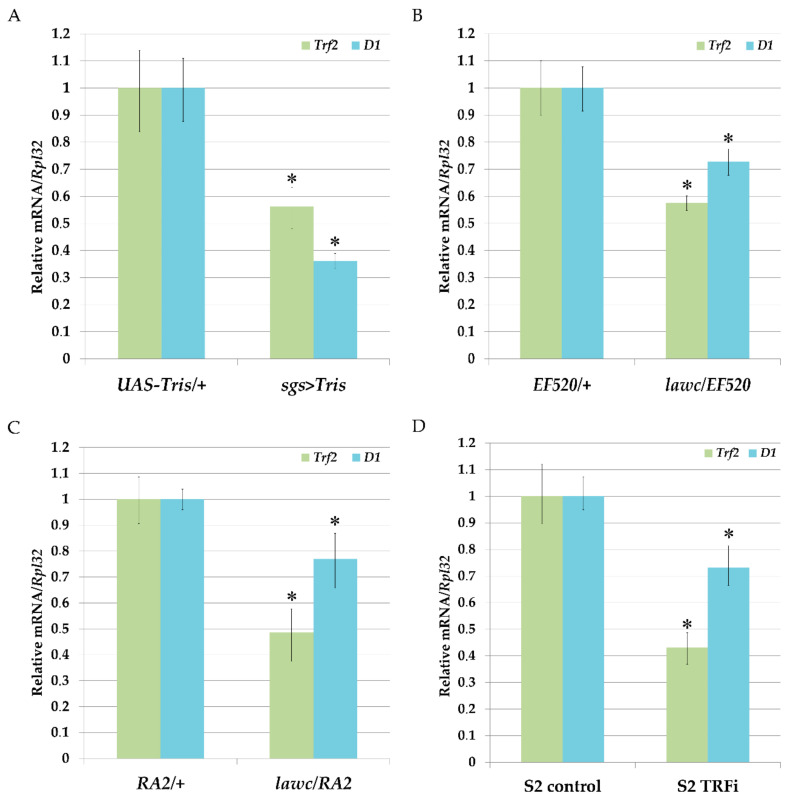
Analysis of *Trf2* and *D1* genes mRNA levels in salivary glands of the third instar larvae, adult flies and S2 cells. The relative expression levels of *Trf2* and *D1* genes were examined using qRT-PCR as described in materials and methods. (**A**) Expression levels of *Trf2* and *D1* genes detected in salivary glands of: control *UAS-Tris/+* larvae and after *Trf2* knock-down by ddRNAi (*sgs* > *Tris*). (**B**) Expression levels of *Trf2* and *D1* genes in *lawc^p1^/EF520* adult flies compared to control *EF520/+* flies. (**C**) Expression levels of *Trf2* and *D1* genes in *lawc^p1^/Df(1)RA2* adult flies compared to control *Df(1)RA2/+* flies; (**D**) expression levels of *Trf2* and *D1* genes in the S2 cells after RNAi-mediated silencing of *Trf2* (S2-TRFi) compared to control (S2 cells treated with dsRNA corresponding to eGFP gene). Data are presented as the mean ± SD of three independent experiments. * *p* < 0.05, compared to control.

**Figure 6 ijms-21-08871-f006:**
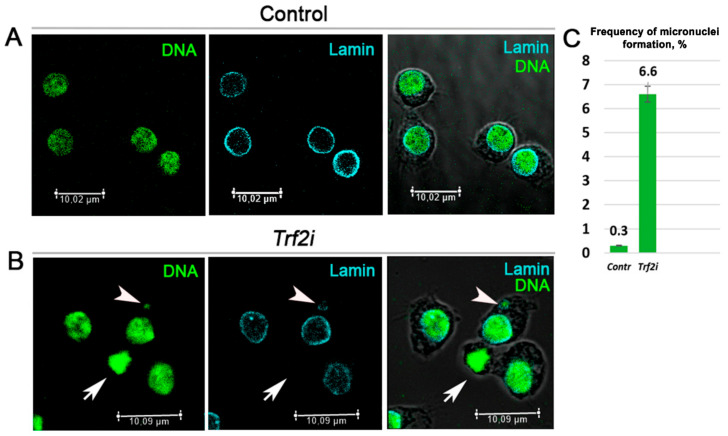
RNAi-mediated depletion of Trf2 leads to micronuclei formation and disturbs nuclear envelope. Confocal immunofluorescence images of *Drosophila* S2 cells (**A**) before and (**B**) after RNAi-mediated silencing of *Trf2*. Cells were stained for SytoxGreen to highlight DNA (green) and anti-Lamin Dm to visualized nuclear envelope (cian). The third column represents merged images. Arrowheads pointe to micronucleus, arrows indicate Lamin negative nuclear envelope. (**C**) Quantification of cells with micronuclei from control (Contr, *n* = 350) and *Trf2* dsRNA transfected cells (*Trf2i*, *n* = 229) from three independent experiments. A *p*-value from Student’s *t*-test was 0.0047. Error bars: SD.

**Table 1 ijms-21-08871-t001:** X-chromosome nondisjunction and existence of attached-X chromosomes in females with lethal *Trf2* mutations. The first column indicates *Trf2* alleles balanced on *In(1)FM* and control stocks. The second column indicates the presence (“+”) or absence (“−”) of females with attached-X chromosomes. The third column indicates the frequency of X-chromosome nondisjunction Qn. The number (N) of analyzed flies in the progeny of females with X-chromosome nondisjunction is given in brackets. The fourth column indicates the frequency of parental females with X-chromosome nondisjunction Qf. The fifth column indicates the number of parental females analyzed.

Stocks	Attached X	Qn, % (N)	Qf, %	Total
*l(1)G0425/In(1)FM*	+	7.17 (779)	40	25
*l(1)G0356/In(1)FM*	+	11.59 (845)	46.15	26
*l(1)G0424/In(1)FM*	−	17.39 (777)	40	25
*l(1)G0295/In(1)FM*	−	14.51 (1201)	48.15	27
*l(1)G0166/In(1)FM*	−	29.6 (236)	20	30
*l(1)G0152/In(1)FM*	−	28.25 (370)	26.66	30
*l(1)EF520/In(1)FM*	−	29.12 (88)	6.67	30
**Control stocks**				
*In(1)FM*	−	1.4 (2445)	1.54	65
*l(1)G0071/In(1)FM*	−	1.5 (650)	1.43	70

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
