# Peer review of "TBP-Related Factor 2 as a Trigger for Robertsonian Translocations and Speciation"

_ijms, 2020, doi:10.3390/ijms21228871_

Round 1

Reviewer 1 Report

The authors present an interesting paper documenting the increase in attached X phenotypes associated with alleles of TBP-related 2 mutants.  They also link the mutants to D1 expression changes.

This work connects to the role of the centromere in chromosomal abberations, which has application in cancer, speciation, and genome architecture research.

Some things could be improved:

1. The first lines of the introduction (28-43) read like a list of bullet points in a powerpoint presentation and not like a written paragraph. These single-sentence "paragraphs" could be more effectively pulled together into one introductory paragraph.

2. The introduction reminds me of two areas of research that seem connected to the authors' work, but which they seem unaware of.  I present some information to my colleagues for their edification, with no expectations that they include related citations here. However, these areas of work seem related to their work and they should be aware of it (whether they include references to the work in this manuscript is up to them).

     2.1. Hybrid incompatibility.  hmr/Lhr research in Drosophila demonstrates that these two genes are involved in hybrid incompatibility, bind chromosome, and specifically, bind centromeres, and seem to serve to suppress transposable element expression on centromeres.  

     2.2. Genome size evolution.  Typically when people study genome architecture they think of either numbers of chromosomes (studied here) and sizes of chromosomes.  There are genome size estimates for the Drosophila DGRP strains, showing tens of Mb in variation in genome size.  Of interest with respect to large genomes: the strains with larger genomes tend to also be associated with larger variation in genome size.  The mechanism described here would possibly explain how this could happen and I suspect there would be clear signs of variation in genome size in the crosses studied here.

3. Related to the above: there is not much attention paid to other ways that chromosomal rearrangements can occur, their relative abundances, etc.

4. In the Results the authors present work on "normal" and "exceptional" genotypes.  Since results come before methods in this work, it is not immediately clear what the authors mean without breaking to read the methods.  A figure may help to demonstrate what the authors mean as the color, eye shape phenotypes will be obvious.  Also - I am not clear on what authors mean by a gray phenotype, so a picture may help readers like me out.

5. Table 1.  What is the number of individuals or crosses associated with the percentages reported?  It would be nice to see in the table.  Also - Why were control stocks not tested for Qf?

6. Line 143 - do authors mean bearing instead of barring?

7. Quarto - do authors mean Quarter? Further, "only a quarto/quarter" or "only quarto/quarter"?

      7.1. This line is a good example of a fairly large number of minor, but in aggregate mildly distracting, grammar errors.  Other inconsistent uses of singular/plural forms of words also occur.  A final read through to correct these will keep from breaking concentration of some readers, which can distract from the point in such sentences.

8. I had to look up what a micronucleus was.  Once I knew, this section makes a lot of sense, but it may help to set this concept up in the introduction (or in the beginning of that section of the paper) more.

Overall, this paper is an interesting contribution to our understanding of the mechanisms that lead to chromosomal fusions.

Author Response

We thank both reviewers for their comments and their advice on how to improve the manuscript. We have now modified the manuscript in accordance with the reviewers’ suggestions and answered all of their concerns. Detailed answers to specific comments are below.

Reviewer #1

1. «The first lines of the introduction (28-43) read like a list of bullet points in a powerpoint presentation and not like a written paragraph. These single-sentence "paragraphs" could be more effectively pulled together into one introductory paragraph»

Answer. We have combined such single-sentences into a paragraph.

2. «The introduction reminds me of two areas of research that seem connected to the authors' work, but which they seem unaware of.  I present some information to my colleagues for their edification, with no expectations that they include related citations here. However, these areas of work seem related to their work and they should be aware of it (whether they include references to the work in this manuscript is up to them).

2.1. Hybrid incompatibility.  hmr/Lhr research in Drosophila demonstrates that these two genes are involved in hybrid incompatibility, bind chromosome, and specifically, bind centromeres, and seem to serve to suppress transposable element expression on centromeres.

Answer. We thank the reviewer for the valuable information. It is highly likely that Trf2 controls hmr/Lhr gene expression. Here, in our study, we did not use interspecific hybrids. However, this may be done in future research.

2.2. Genome size evolution.  Typically when people study genome architecture they think of either numbers of chromosomes (studied here) and sizes of chromosomes.  There are genome size estimates for the Drosophila DGRP strains, showing tens of Mb in variation in genome size.  Of interest with respect to large genomes: the strains with larger genomes tend to also be associated with larger variation in genome size.  The mechanism described here would possibly explain how this could happen and I suspect there would be clear signs of variation in genome size in the crosses studied here. »

Answer. We believe that our research will form the basis for a detailed study of the role of Trf2 in the creation of a transcription program responsible for the structuring of pericentromeric regions and, in particular, will bring a closer understanding of the problem of genome size evolution.

3. «Related to the above: there is not much attention paid to other ways that chromosomal rearrangements can occur, their relative abundances, etc».  

Answer. We tried to focus on specific points that our research is devoted to: centric fusion, pericentromeric regions, chromocenter and Trf2.

4. «In the Results the authors present work on "normal" and "exceptional" genotypes.  Since results come before methods in this work, it is not immediately clear what the authors mean without breaking to read the methods.  A figure may help to demonstrate what the authors mean as the color, eye shape phenotypes will be obvious.  Also - I am not clear on what authors mean by a gray phenotype, so a picture may help readers like me out».

Answer. We took into account the wishes of the reviewer, drew a picture and inserted it into the manuscript (see Figure 1).

5. Table 1.  What is the number of individuals or crosses associated with the percentages reported?  It would be nice to see in the table.  Also - Why were control stocks not tested for Qf?

Answer. We added the number of analyzed flies to Table 1. The control frequency of parental females with nondisjunction (Qf) was rather low. We added it to Table 1 and to the histogram. 

6. «Line 143 - do authors mean bearing instead of barring? »

Answer. It was a misprint. It is improved.

7. «Quarto - do authors mean Quarter? Further, "only a quarto/quarter" or "only quarto/quarter"?

7.1. This line is a good example of a fairly large number of minor, but in aggregate mildly distracting, grammar errors.  Other inconsistent uses of singular/plural forms of words also occur.  A final read through to correct these will keep from breaking concentration of some readers, which can distract from the point in such sentences»

Answer. We mean ¼. We improved “quarto” to “a quarter”.

8. «I had to look up what a micronucleus was.  Once I knew, this section makes a lot of sense, but it may help to set this concept up in the introduction (or in the beginning of that section of the paper) more»

Answer. Micronucleus formation is a D1 mutation phenotype. We used this fact as confirmation that Trf2 can control the expression of D1 and possibly other genes responsible for the formation of pericentromeric regions.

Reviewer 2 Report

The manuscript by Cherezov et al. describe an elegant study aimed at the demonstration of the link between the Trf2 expression and the organization of the pericentromeric regions in Drosophila.
The text is overall well written and the detailed description of the genetic approach would be beneficial for students that will read the paper.
However, I have concerns on how some experiments have been performed. Please, find below all my comments (both major and minor).

The introduction could benefit of a description of the heterochromatin in Drosophila which has shown peculiar features if compared to other organisms (a comprehensive review is provided in Marsano et al., 2019 doi:10.1016/j.tig.2019.06.002)

l48 "Obviously, such mutants will have a high frequency of chromosome non-disjunction" this is not ever true. Heterozygous robertsonian translocations give rise to a complex meiotic figures (tri- and tetravalents) that allow the segregation of chromosomes. This are not properly non-disjunction events

l 57-58 are there evidence concerning the link between the heterochromatin organization and the chance to get a centric fusion? or is this an Authors' hypothesis?

l59 "One way of eukaryotic genome organization is chromocenter" This statement is misleading for naive readers. please cosider rewording to explain the notion that only part of the genome can organize the chromocenter.

l69 "main peculiarity" do you mean main phenotype?

l78 "high conservatism" should be ubiquitous presence

l78 "on" should be in

l90 "predicted" hypothesized

l92 "a new genetic screening" should be new genetic screenings

l11 "an attached" should be attached X chromosomes

l113-114 "In crosses of normal males to females carrying an attached X, male progeny inherit their X from their father and their Y from their mother 114 (X^X/Y)" This is incorrect. Y comes from males. By the way, XXY develop as females.

l184 "The similar results..." should be "similar results"

l198 RNAi-silencing. This experiment seems to lack a proper control. While the untransfected S2 cells represent an additional experimental control, the real control should contain either the scrambled Trf2 sequence used in the experiment or at least the dsRNA directed against GFP, as designed for the in vivo experiments.

Figure 4C Friquence should be frequency

l203 Figure 4 legend. sels should be cells

l 300 "....XX/X super-females also die" this statement is incorrect. Bridges observed a sub-lethal phenotype associated to metafemales. There is also no reason for expecting complete lethality from this crosses. Please, explain or revise.

l326 witch should be which

l397-399 More details should be provided concerning the transgenic line expressing the dsRNA. Was the insertion mapped? Was the transgene expression measured?

It is not clear from the text if the presence of attached X chromosomes has been evaluated cytologically (e.g. from metaphase chromosomes preparation). This should be done since all the work is based on the assumption that attached Xs are generated.

Also, why the Rpl32 and Trf2 gene expression were assayed using the Taqman chemistry while the D1 expression was assayed using the SYBR-based chemistry? How can the normalization be done if the two values have been obtained with different approaches? Is this data reliable? Can be the expression of the Trf1 and D1 compared? I have concerns about this kind of approach.

Figure 3 show the results of the expressiona anlyses in strains in which Trf2 has been downregulated. Opposite results were obtained in salivary glands and in other experimental conditions (figure 3B anc C) . On the contrary, the Authors state that they obtained similar results when the lawcp1/l(1)EF520 and lawcp1/Df(1)RA2 mutant flies were examined, compared to the expression in salivary glands. The Authors should rebiult this paragraph and correctly comment on the results obtained. The conclusions that have been drawn in the current version of the manuscript should be also carefully reconsidered.

Author Response

We thank both reviewers for their comments and their advice on how to improve the manuscript. We have now modified the manuscript in accordance with the reviewers’ suggestions and answered all of their concerns. Detailed answers to specific comments are below.

Reviewer #2

«The introduction could benefit of a description of the heterochromatin in Drosophila which has shown peculiar features if compared to other organisms (a comprehensive review is provided in Marsano et al., 2019 doi:10.1016/j.tig.2019.06.002).»

Answer. We tried to focus on specific points that our research is devoted to: centric fusion, pericentromeric regions, chromocenter and Trf2. We will use the suggested link in future publications.

«l48 "Obviously, such mutants will have a high frequency of chromosome non-disjunction" this is not ever true. Heterozygous robertsonian translocations give rise to a complex meiotic figures (tri- and tetravalents) that allow the segregation of chromosomes. This are not properly non-disjunction events»

Answer. We replaced "Obviously, such mutants will have a high frequency of chromosome non-disjunction" to Obviously, such mutants will have an increased rate of chromosome non-disjunction compared to karyotypically normal animal”.

«l57-58 are there evidence concerning the link between the heterochromatin organization and the chance to get a centric fusion? or is this an Authors' hypothesis?»

Answer. It is our hypothesis.

«l59 "One way of eukaryotic genome organization is chromocenter" This statement is misleading for naive readers. please cosider rewording to explain the notion that only part of the genome can organize the chromocenter

Answer. We have made a correction. An explanation of this statement is given in the following sentences in the same paragraph below.

«l69 "main peculiarity" do you mean main phenotype?»

Answer. We mean that a high frequency of chromosome nondisjunction is the main peculiarity of mutations that disrupt the chromocenter. We improved this: “In screening for genes that control the formation and reorganization of chromocenter, the high frequency of chromosome nondisjunction is the main peculiarity in the progeny of mutant parents.”

«l78 "high conservatism" should be ubiquitous presence»

Answer. We improved this

«l78 "on" should be in»

Answer. We improved this

«l90 "predicted" hypothesized»

Answer. We improved this

«l92 "a new genetic screening" should be new genetic screenings»

Answer. We improved this

«l11 "an attached" should be attached X chromosomes»

Answer. We improved this

«l113-114 "In crosses of normal males to females carrying an attached X, male progeny inherit their X from their father and their Y from their mother 114 (X^X/Y)" This is incorrect. Y comes from males. By the way, XXY develop as females»

Answer. Precisely because XXY flies develop as females, the male progeny inherit their Y from X^X/Y mother, so our proposal is correct.

«l184 "The similar results..." should be "similar results

Answer. We improved this

«l198 RNAi-silencing. This experiment seems to lack a proper control. While the untransfected S2 cells represent an additional experimental control, the real control should contain either the scrambled Trf2 sequence used in the experiment or at least the dsRNA directed against GFP, as designed for the in vivo experiments»

Answer. The S2 cells demonstrated in Figure 4 were treated the same way as the S2 cells in Figure 3. Control cells were transfected by dsRNA corresponding to eGFP gene in both cases.

«Figure 4C Friquence should be frequency»

Answer. We improved this.

«l203 Figure 4 legend. sels should be cells»

Answer. We improved this

l 300 "....XX/X super-females also die" this statement is incorrect. Bridges observed a sub-lethal phenotype associated to metafemales. There is also no reason for expecting complete lethality from this crosses. Please, explain or revise.

Answer. Super-females had low viability and died at an early age. Therefore, we did not consider them. For example see Fig 12 and 14 in: C. Kaufman “A Short History and Description of Drosophila melanogaster Classical Genetics: Chromosome Aberrations, Forward Genetic Screens, and the Nature of Mutations” GENETICS June 1, 2017 vol. 206 no. 2 665-689; https://doi.org/10.1534/genetics.117.199950 or: https://bdsc.indiana.edu/stocks/aberration/compound_x_overview.html#:~:text=Compound%20or%20attached%2DX%20chromosomes,their%20Y%20from%20their%20mother.

«l326 witch should be which»

Answer. We improved this

«l397-399 More details should be provided concerning the transgenic line expressing the dsRNA. Was the insertion mapped? Was the transgene expression measured

Answer. Using inverse PCR we found only one copy of UAS-Tris transgenic construct that was located 539 bp upstream of CG4781 5’-termini on chromosome 2R. The red arrow indicates the position of UAS-Tris insertion in the map (see Figure attached).

The expression levels of CG4781 were not changed upon insertion and overexpression of Trf2-knockdown transgene as evaluated by RT-PCR in preliminary experiments. UAS-mediated inducible overexpression of Trf2 knock-down transgene was confirmed using RT-PCR with primer for hsp70 promoter located in pUAST vector and one of primers used for Trf2 knock-down transgene construction (Trf2XbaI) (see figure below). Total RNA for evaluating of transgene expression was extracted from salivary glands of UAS-Tris/+ (control) and sgs-Gal4>Uas-Tris third instar larvae.

To corroborate that observed phenotype after TRF2 knock-down was caused only by TRF2 itself, we performed rescue experiment using Drosophila line overexpressing TRF2 protein as described in our earlier work [Cherezov et al., 2013]. https://pubmed.ncbi.nlm.nih.gov/23789418/

«It is not clear from the text if the presence of attached X chromosomes has been evaluated cytologically (e.g. from metaphase chromosomes preparation). This should be done since all the work is based on the assumption that attached Xs are generated».

Answer. The nature of the inheritance of traits implies the attached X chromosomes. However, we used the method of preparing metaphase chromosome plates and demonstrated the attached X chromosomes cytologically (see new Fig. 2).

«Also, why the Rpl32 and Trf2 gene expression were assayed using the Taqman chemistry while the D1 expression was assayed using the SYBR-based chemistry? How can the normalization be done if the two values have been obtained with different approaches? Is this data reliable? Can be the expression of the Trf1 and D1 compared? I have concerns about this kind of approach».

Answer. In our study, we determined the differences in gene expression levels using the relative quantification (2-ΔΔCt) method. For the 2-ΔΔCt calculation to be valid, the amplification efficiencies of the target and reference must be approximately equal (Livak, Schmittgen, 2001). The effect on the target–reference ratio depends on the PCR efficiency of the target and reference amplicons (https://pubmed.ncbi.nlm.nih.gov/20138998/). In our preliminary experiments using standard curve obtained with 2-fold cDNA dilution series, we determined the efficiency of amplification for D1 primer pair and for Trf2 and Rpl32 primer pairs-probe combinations. We found that the efficiencies of amplification with D1 primer pair and with Trf2 and Rpl32 primer pairs-probes were near the same (96%, 95% and 98%, respectively). We also verified that chosen D1 primer pair did not produce dimers as assessed by melting curve analysis and did not amplify any undesired amplicons as assessed by size using PAGE. The use of SYBR-based chemistry for TRF2 and Rpl32 was not possible because we were not able to pick-up primers pair that would not produce many primer dimers during amplification, so were forced to use TaqMan-based chemistry for these genes. Applied biosystems 7500 software v.2.3 used in our study allows to use different chemistries in one assay for different genes and to take into consideration the efficiency of amplification for all primer pairs and primer pair-probe combinations. Both, the TaqMan chemistry and SYBR-based chemistry, reflect the real levels of mRNA. By optimization of SYBR Green method, its performance and quality could be comparable to TaqMan method (https://www.ncbi.nlm.nih.gov/pmc/articles/PMC3988599/, https://pubmed.ncbi.nlm.nih.gov/18620571/). The main difference between TaqMan probe-based chemistry and SYBR Green I dye chemistry is that SYBR-based chemistry if not well-optimized could detect non-specific amplification products, but we used optimized primer pair. There are no restrictions for use of different types of chemistries for target and reference genes in one assay and for calculation of differences in gene expression using the Comparative Ct method (2-ΔΔCt) used in “Applied Biosystems 7500 software” because all primer pairs and primer pair-probe combination were stable and validated and their amplification efficiencies were near identical and taken into consideration during calculation. In our study, we are not comparing expression levels of Trf2 to D1 or vice versa, instead we are analyzing the changes of normalized expression in experimental samples compared to control for each gene separately using relative quantification (comparative Ct method or 2-ΔΔCt) method implemented in Applied biosystems 7500 software.

«Figure 3 show the results of the expressiona anlyses in strains in which Trf2 has been downregulated. Opposite results were obtained in salivary glands and in other experimental conditions (figure 3B anc C) . On the contrary, the Authors state that they obtained similar results when the lawcp1/l(1)EF520 and lawcp1/Df(1)RA2 mutant flies were examined, compared to the expression in salivary glands. The Authors should rebiult this paragraph and correctly comment on the results obtained. The conclusions that have been drawn in the current version of the manuscript should be also carefully reconsidered».

Answer. We considered results obtained in salivary glands similar to that obtained in lawcp1/l(1)EF520 and lawcp1/Df(1)RA2 mutant flies because there is a correlation between Trf2 and D1 expression levels, i.e. D1 expression is downregulated upon decreased TRF2 gene expression, although the degree of Trf2 and D1 genes downregulation varies in different experimental conditions.

Round 2

Reviewer 2 Report

The Authors have sufficiently improved the manuscript.

I would like to suggest some additional minor modification before acceptance.

«l57-58 are there evidence concerning the link between the heterochromatin organization and the chance to get a centric fusion? or is this an Authors' hypothesis?»
Answer. It is our hypothesis.
My comment: In this case it should be clearly stated in the text

«l397-399 More details should be provided concerning the transgenic line expressing the dsRNA. Was the insertion mapped? Was the transgene expression measured?»
Answer. Using inverse PCR we found .....

My comment.  Can be this paragraph provided as supplementary information?

"By optimization of SYBR Green method, its performance and quality could be comparable to TaqMan method (https://www.ncbi.nlm.nih.gov/pmc/articles/PMC3988599/, https://pubmed.ncbi.nlm.nih.gov/18620571/). "

My comment. Please, provide this explanation in the methods' section along with these references, to justify the experimental design.

Author Response

Answers to the reviewers’ comments

«l57-58 are there evidence concerning the link between the heterochromatin organization and the chance to get a centric fusion? or is this an Authors' hypothesis?»
Answer. It is our hypothesis.
My comment: In this case it should be clearly stated in the text

Answer:

We have changed the sentence in the text (l53-54) to “We believe that it is more likely that mutations in genes that disrupt the organization of pericentromeric regions will promote centric fusion of chromosomes”.

«l397-399 More details should be provided concerning the transgenic line expressing the dsRNA. Was the insertion mapped? Was the transgene expression measured?»
Answer. Using inverse PCR we found .....

My comment.  Can be this paragraph provided as supplementary information?

Answer:

We have provided this information in Supplementary Figure 2.

"By optimization of SYBR Green method, its performance and quality could be comparable to TaqMan method (https://www.ncbi.nlm.nih.gov/pmc/articles/PMC3988599/, https://pubmed.ncbi.nlm.nih.gov/18620571/). "

My comment. Please, provide this explanation in the methods' section along with these references, to justify the experimental design.

Answer:

We have provided this information in Materials and Methods.